# A Robust Disturbance Rejection Whole-Body Control Framework for Bipedal Robots Using a Momentum-Based Observer

**DOI:** 10.3390/biomimetics10030189

**Published:** 2025-03-19

**Authors:** Shuai Heng, Xizhe Zang, Yan Liu, Chao Song, Boyang Chen, Yue Zhang, Yanhe Zhu, Jie Zhao

**Affiliations:** State Key Laboratory of Robotics and System, Harbin Institute of Technology, Harbin 150080, China; zangxizhe@hit.edu.cn (X.Z.); liuyan1631998@163.com (Y.L.); 20b908062@stu.hit.edu.cn (C.S.); 20b908061@stu.hit.edu.cn (Y.Z.); yhzhu@hit.edu.cn (Y.Z.); jzhao@hit.edu.cn (J.Z.)

**Keywords:** biped robot, divergent component of motion, momentum-based observer, whole-body control

## Abstract

This paper presents a complete planner and controller scheme for bipedal robots, designed to enhance robustness against external disturbances. The high-level planner utilizes model predictive control (MPC) to optimize both the foothold location and step duration based on the divergent component of motion (DCM) to increase the robustness of generated gaits. For low-level control, we employ a momentum-based observer capable of estimating external forces acting on both stance and swing legs. The full-body dynamics, incorporating estimated disturbances, are integrated into a weighted whole-body control (WBC) to obtain more accurate ground reaction forces needed by the momentum-based observer. This approach eliminates the dependency on foot-mounted sensors for ground reaction force measurement, distinguishing our method from other disturbance estimation methods that rely on direct sensor measurements. Additionally, the controller incorporates trajectory compensation mechanisms to mitigate the effects of external disturbances. The effectiveness of the proposed framework is validated through comprehensive simulations and experimental evaluations conducted on BRUCE, a miniature bipedal robot developed by Westwood Robotics (Los Angeles, CA, USA). These tests include walking under swing leg disturbances, traversing uneven terrain, and simultaneously resisting upper-body pushes.

## 1. Introduction

Legged robots, owing to their exceptional adaptability to complex terrains, demonstrate significant potential for diverse practical applications. In recent years, legged robots like quadrupeds and bipeds have received considerable research attention. Quadrupedal robots, for instance, Boston Dynamics’ Spot and Unitree’s B2, have successfully been deployed in industrial environments. However, the practical implementation of bipedal robots remains limited. During real-world operations, bipedal robots are inevitably subjected to various external disturbances, including impulsive forces, environmental collisions, and dynamic load variations. These disturbances may lead to trajectory deviations and, in severe cases, loss of stability. As a result, the development of robust disturbance rejection capabilities is crucial for ensuring stable bipedal locomotion in practical applications.

To date, most model-based locomotion control strategies are implemented by solving an optimal control problem [1]. However, because the bipedal system is high-dimensional, hybrid, nonlinear, and highly constrained, solving this problem in real time remains computationally intractable. Consequently, most approaches break down the problem into multiple stages, employing either abstract models or full dynamic models at each stage. While this hierarchical approach sacrifices global optimality, it enables real-time implementation [1,2,3]. A widely adopted framework employs a two-level hierarchical structure, where the high-level plans the footsteps and the low level optimizes the contact forces [4,5,6].

The high-level planner aims to generate trajectories in real-time, usually adopting abstract models such as the linear inverted pendulum model (LIPM) [7,8]. Kajita et al. [9] developed a preview control approach to generate center of mass (CoM) trajectory. Their method solved a discrete-time, infinite-horizon linear quadratic regulator (LQR) problem based on LIPM. Wieber [10] enhanced the preview control by using a linear MPC scheme, which is capable of handling inequality constraints, including stability bounds and reachability limits. The MPC and LIPM frameworks can be extended to generate other trajectories, like footstep [11] and CoM height trajectories [12]. Further developments revealed that the LIPM dynamics can be divided into stable and unstable components. Therefore, the controller can only consider the unstable parts, known as DCM [13]. Additionally, researchers recognized the potential benefits of variable step durations. Khadiv et al. [14,15] demonstrated this through their work on simultaneous optimization of landing positions and step durations using the DCM model, showing that timing adjustment can enhance robustness.

For the low-level controller, there has been a convergence among many researchers working with torque-controlled bipeds, formulating the controller as a quadratic program (QP) [3,4,16,17]. This QP formulation optimizes joint torques, joint accelerations, contact wrenches, and angular momentum simultaneously, enabling the execution of multiple tasks while satisfying system dynamics and contact force constraints. The key advantage of this method is that it provides a strict hierarchy, which guarantees each tasks resolved in order. Depending on how the task hierarchy is managed, QP-based WBC can be classified into two main categories. The weighted WBC scheme sets all operational tasks as objectives of a single QP with priorities implicitly being enforced with weights [3]. The null space projection WBC method achieves a strict task hierarchy by projecting the lower priority tasks into the null space of higher priority tasks [18].

Robustness against external disturbances in legged robots has been extensively studied in recent years. Current approaches can be broadly categorized into two main types. The first category relies on state feedback to adjust control strategies, including joint-specific torque compensation [19,20], foot force planning [21,22], and support configuration adjustment through step-taking [23,24]. However, these methods lack precise external disturbance estimation; therefore, their effectiveness in resisting larger external disturbances is limited [25]. The second category employs momentum-based observer to estimate external disturbance. For example, Englsberger et al. [26] developed a momentum-based disturbance observer, enabling the TORO robot to withstand external disturbances equivalent to 15% of its body weight in simulations. Focchi et al. [27] further extended this algorithm to angular cases by estimating the complete wrench at CoM and implemented it on the HyQ quadruped robot. Nevertheless, these momentum-based methods typically require additional force sensors mounted on the robot’s feet, making them incompatible with many robotic platforms such as BRUCE.

With respect to the literature review depicted above, this paper aims to proposes a novel control framework that addresses two critical limitations in existing approaches: (1) the dependency on force sensors for external disturbance resistance and (2) the lack of integration between robust planning and control methods. The proposed framework comprises three key components. First, a high-level planner utilizes MPC to optimize both the foothold location and step duration based on the DCM to enhance the robustness of generated gaits. Second, a momentum-based observer estimates disturbances acting on both stance and swing legs. Third, a weighted WBC incorporates full-body dynamics with the estimated disturbances to compute more accurate ground reaction forces required for the observer. Moreover, WBC compensates for the estimated disturbance effects on each task.

Zhu et al. [25] also employed a disturbance observer to estimate external disturbance without requiring additional force sensors on the robot’s feet and performed online disturbance compensation within the WBC. However, their method only estimates the external disturbance on CoM, which is similar to estimating disturbances on the stance leg [28]. While estimating external wrenches acting solely on the CoM can improve locomotion robustness on uneven terrain, it fails to prevent the robot from falling after severe impact to the swing leg. Morlando et al. [28] estimated disturbances for both stance and swing legs and compensated for external forces acting on both legs. However, their method requires force sensors on the robot’s feet. In addition, we adopt a different planning method, which use MPC to plan both step locations and durations, further enhancing robustness.

This paper is organized as follows. Section 2 presents the mathematical derivation of the momentum-based observer. Section 3 introduces whole control scheme, including the footstep planner and weighted WBC method. Section 4 presents the simulation and experimental results of the control scheme applied on BRUCE. In the end, conclusions and discussion are provided in Section 5.

## 2. Momentum-Based Observer for Disturbance Estimation

In this section, the dynamics model of a biped robot is first presented for momentum-based observer design purposes. Afterward, the momentum-based observer, which estimates disturbances acting on both the stance and swing legs, is introduced.

### 2.1. Dynamic Model

To effectively demonstrate the impact of external forces perceived by the momentum-based observer, we establish a floating base dynamics model for bipedal robots. This model typically consists of a free-floating base with two legs, which effectively captures the influence of external forces on both the support leg and the swing leg. Let {C} denote the frame attached to the robot’s CoM. The free-floating base is connected to the fixed world frame {W} through six virtual joints, resulting in six degrees of freedom. Virtual joints, unlike the joints in the legs, do not physically exist and have no actual rotational axes. They are introduced to describe the position and posture of the robot’s upper body, allowing the joint values to serve as generalized coordinates for describing the robot’s upper body configuration. A frame {B} is attached to the floating base to represent its position and orientation. Additionally, two legs are connected to the floating base, adding 2n degrees of freedom to the bipedal robot, where n>0 represents the number of joints in each leg. Figure 1 illustrates the biped robot and its corresponding coordinate frames.

Let $x_{b}={\left[x_{b},y_{b},z_{b}\right]}^{T}\in R^{3}$, ${\dot{x}}_{b},{\ddot{x}}_{b}\in R^{3}$ denote the position, velocity, and acceleration of the origin of frame {B} relative to the frame {W}, respectively. Similarly, $\omega_{b},{\dot{\omega }}_{b}\in R^{3}$ represent the angular velocity and angular acceleration of frame {B} relative to frame {W}, respectively. Additionally, variables are defined to describe the position of CoM: $x_{c}={\left[x_{c},y_{c},z_{c}\right]}^{T}\in R^{3}$, ${\dot{x}}_{c},{\ddot{x}}_{c}\in R^{3}$ denote the position, velocity, and acceleration of the origin of frame {C} relative to {W}, respectively. $\omega_{c},{\dot{\omega }}_{c}\in R^{3}$ are the angular velocity and angular acceleration of frame {C} relative to {W}, respectively. The orientation of frame {B} relative to {W} is described by the rotation matrix $R_{b}\in \mathrm{SO}\left(3\right)$, from which the ZYX Euler angles $\phi \in R^{3}$ can be derived.

When the floating base frame {B}’s coordinates and joint space coordinates are utilized as generalized coordinates, the dynamics of the robot can be expressed as follows:(1)$$\bar{M}\left(\begin{array}{c} {\ddot{x}}_{b}\\ {\dot{\omega }}_{b}\\ \ddot{q} \end{array}\right)+\bar{C}\left(\begin{array}{c} {\dot{x}}_{b}\\ \omega_{b}\\ \dot{q} \end{array}\right)+\bar{G}=\left(\begin{array}{c} 0\\ 0\\ \tau_{\mathrm{joint}} \end{array}\right)+\tau_{\mathrm{ee}}+\tau_{\mathrm{ext}}$$
where $\bar{M}$, $\bar{C}$, and $\bar{G}$ are the inertia matrix, Coriolis and centrifugal force matrix, and gravity matrix, respectively, all expressed in frame {B}. $q,\dot{q},\ddot{q}\in R^{2n}$ represent the joint positions, velocities, and accelerations of the legs, respectively. $\tau_{\mathrm{joint}}\in R^{2n}$ denotes the joint torques of the legs, $\tau_{\mathrm{ee}}$ represents the generalized external forces due to ground reaction forces on the support leg, and $\tau_{\mathrm{ext}}$ represents the generalized external forces due to external disturbances.

For $\tau_{\mathrm{ee}}$, it can be expressed as follows:(2)$$\tau_{\mathrm{ee}}=\sum_{i=1}^{n_{ee}}\left[\begin{array}{c} \left[\begin{array}{cc} I & 0\\ {\hat{x}}_{bi} & I \end{array}\right]\\ {\bar{J}}_{i}^{T} \end{array}\right]F_{i}$$
where $n_{ee}$ is the number of contact end-effectors. The bipedal robot has two feet as contact end-effectors, so $n_{ee}=2$. ${\bar{J}}_{i}$ is the Jacobian matrix of the contact end-effector with dimensions $R^{6\times (2n+6)}$. $x_{bi}$ is the offset of the end-effector frame ${\left\{T\right\}}_{i}$ relative to frame {B}, given by $x_{bi}=x_{i}-x_{b}$. The hat operator can turn a vector $a\in R^{3}$ into a 3×3 skew symmetric matrix such that $\hat{a}b=a\times b\;\forall b\in R^{3}$. $F_{i}$ is the wrench applied to the contact end-effector.

According to [29], using the position and orientation of the CoM frame as generalized coordinates, as opposed to those of the base frame, simplifies the dynamics model of the floating base robot and facilitates the direct control of the CoM. The transformation between the generalized coordinates using the CoM frame’s coordinates and those using the floating base frame’s coordinates is given by the following:(3)$$\left(\begin{array}{c} {\dot{x}}_{c}\\ \omega_{c}\\ \dot{q} \end{array}\right)=\left[\begin{array}{ccc} I & -{\hat{x}}_{bc} & J_{bc}\\ 0 & I & 0\\ 0 & 0 & I \end{array}\right]\left(\begin{array}{c} {\dot{x}}_{b}\\ \omega_{b}\\ \dot{q} \end{array}\right)=\bar{T}\left(\begin{array}{c} {\dot{x}}_{b}\\ \omega_{b}\\ \dot{q} \end{array}\right)$$
where $x_{bc}$ is the relative position deviation of the CoM with respect to the frame {B} expressed in frame {W}, and $J_{bc}$ is the Jacobian matrix of the position deviation $x_{bc}$.

Thus, the dynamics equation expressed with the CoM frame’s coordinates, as in [30], is obtained as follows:(4)$$\underset{M}{\underset{︸}{\left[\begin{array}{ccc} mI & 0 & 0\\ 0 & M_{11} & M_{12}\\ 0 & M_{21} & M_{22} \end{array}\right]}}\underset{\dot{\nu }}{\underset{︸}{\left(\begin{array}{c} {\ddot{x}}_{c}\\ {\dot{\omega }}_{c}\\ \ddot{q} \end{array}\right)}}+\underset{C}{\underset{︸}{\left[\begin{array}{c} C_{1}\\ C_{2}\\ C_{3} \end{array}\right]}}\underset{\nu }{\underset{︸}{\left(\begin{array}{c} {\dot{x}}_{c}\\ \omega_{c}\\ \dot{q} \end{array}\right)}}+\underset{G}{\underset{︸}{\left[\begin{array}{c} mg\\ 0\\ 0 \end{array}\right]}}=\left(\begin{array}{c} 0\\ 0\\ \tau_{\mathrm{joint}} \end{array}\right)+{\bar{T}}^{-T}\left(\tau_{\mathrm{ee}}+\tau_{\mathrm{ext}}\right)$$

The transformed inertia matrix and Coriolis matrix are given by $M={\bar{T}}^{-T}\bar{M}{\bar{T}}^{-1}$ and $C={\bar{T}}^{-T}\bar{C}{\bar{T}}^{-1}+{\bar{T}}^{-T}\bar{M}{\dot{\bar{T}}}^{-1}$, respectively. The scalar *m* represents the total mass of the bipedal robot, and $g\in R^{3}$ is the gravitational acceleration vector. The first row of the equation corresponds to Newton’s law of motion, which governs the translational dynamics of the system’s CoM. The second row describes the rotational dynamics of the CoM frame, establishing the relationship between its angular acceleration and external forces. The third row models the internal motion of the multi-body system, specifically formulating the dynamics of the leg joint accelerations $\ddot{q}$ [30]. When the leg inertia is small, $M_{12}$, $M_{21}$, $C_{1}$, and $C_{2}$ can be approximated as zero, meaning the Coriolis and centrifugal forces, due to the coupling between the leg motion and the CoM, can be neglected [28].

The simplified dynamics equation is expressed as follows:(5)$$\left[\begin{array}{cc} M_{\mathrm{com}} & 0_{6\times 2n}\\ 0_{2n\times 6} & M_{q} \end{array}\right]\left(\begin{array}{c} {\ddot{x}}_{c}\\ {\dot{\omega }}_{c}\\ \ddot{q} \end{array}\right)+\left[\begin{array}{c} 0\\ 0\\ C_{q} \end{array}\right]+\left[\begin{array}{c} mg\\ 0\\ 0 \end{array}\right]=S^{T}\tau +J_{\mathrm{st}}^{T}f_{\mathrm{gr}}+J_{\mathrm{ext}}^{T}f_{\mathrm{ext}}$$
where $S=[0_{2n\times 6}\;I_{2n}]$ is the selection matrix for joint torques. $J_{\mathrm{st}}\left(q\right)=\left[J_{\mathrm{st},\mathrm{com}}\left(q\right)\;J_{\mathrm{st},j}\left(q\right)\right]\in R^{\left(3\times n_{c}\right)\times \left(6+2n\right)}$ is the Jacobian matrix of the ground reaction force application points, mapping the ground reaction forces to the accelerations of the CoM and leg joints. Here, $0\leq n_{c}\leq 4$ is the number of contact points. For the BRUCE robot, each foot has two contact points (toe and heel), resulting in a maximum of four contact points. $f_{\mathrm{gr}}$ represents the ground reaction forces. $J_{\mathrm{ext}}\left(q\right)=\left[J_{\mathrm{ext},\mathrm{com}}\left(q\right)\;J_{\mathrm{ext},j}\left(q\right)\right]\in R^{\left(3\times 4)\times (6+2n\right)}$ is the Jacobian matrix of the external disturbance application points, mapping the external disturbances to the accelerations of the CoM and leg joints. $f_{\mathrm{ext}}={\left[f_{\mathrm{sw},\mathrm{ext}}^{T},\;f_{\mathrm{st},\mathrm{ext}}^{T}\right]}^{T}$ represents the external disturbances, including those acting on the support leg and swing leg. The last term can be expressed as $J_{\mathrm{ext}}^{T}f_{\mathrm{ext}}={\left[J_{\mathrm{sw}}^{T}\left(q\right)\;J_{\mathrm{st}}^{T}\left(q\right)\right]}^{T}f_{\mathrm{ext}}$.

### 2.2. Momentum-Based Observer Design

The proposed observer utilizes the momentum of both the support leg and the swing leg to estimate external disturbance on both legs. Our method is different from [27], which used only the momentum at the CoM to estimate external disturbances acting on the CoM. While estimating external wrenches acting only on the CoM can enhance locomotion robustness on uneven terrain, it fails to prevent the robot from falling after severe impact to the swing leg. Such impacts can cause the swing foot to deviate from its desired trajectory, potentially leading to missed footholds or even complete failure to touch the ground, resulting in a fall. Thus, our observer addresses a broader range of scenarios compared to methods focused solely on CoM disturbances.

During the dynamic modeling process, the dynamics equations (Equation (Equation 5)) of the robot’s CoM and leg joints have been decoupled, allowing the generalized momentum of the legs to be separately derived as follows:(6)$$k=M_{q}\dot{q}$$

Based on the fundamental properties of the robot’s inertia matrix [31], we have the following:(7)$${\dot{M}}_{q}=C_{q}^{T}+C_{q}$$

Differentiating Equation (Equation 6) and combining it with Equation (Equation 7), then substituting into Equation (Equation 5), yields the time derivative of the generalized momentum as follows:(8)$$\dot{k}=C_{q}^{T}\dot{q}+\tau +J_{\mathrm{st},j}^{T}f_{\mathrm{gr}}+J_{\mathrm{ext},j}^{T}f_{\mathrm{ext}}$$

Let $\hat{f}$ represent the estimate of $f_{\mathrm{ext}}$. Then, $\hat{F}=J_{\mathrm{ext},j}^{T}\hat{f}$ and $F_{\mathrm{ext}}=J_{\mathrm{ext},j}^{T}f_{\mathrm{ext}}$ denote the estimated and actual generalized joint torques due to external disturbances, respectively. The estimated generalized momentum is given by the following:(9)$$\hat{k}\left(t\right)=\int_{0}^{t}\underset{\dot{\hat{k}}}{\underset{︸}{C_{q}^{T}\dot{q}+\tau +J_{\mathrm{st},j}^{T}f_{\mathrm{gr}}+\hat{F}\left(\sigma \right)}}d\sigma$$

Here, the initial estimated generalized momentum is assumed to be zero. Then, the estimated generalized external force is derived as follows:(10)$$\hat{F}=K_{1}\left(k\left(t\right)-\int_{0}^{t}\dot{\hat{k}}\left(\sigma \right)d\sigma \right)$$
where $K_{1}$ is a positive definite diagonal matrix containing the gains. Combining Equations (Equation 8) and (Equation 10), the observer dynamics can be expressed as follows:(11)$$\dot{\hat{F}}=K_{1}\left(F_{\mathrm{ext}}-\hat{F}\right)$$

According to [32], a second-order dynamic model can better attenuate the effects of high-frequency noise (e.g., introduced by both the IMU sensor and the joint velocity measurements), thereby improving the observer’s estimation accuracy. To achieve more accurate external force estimation, we extended this first-order dynamic model to an arbitrary-order observer dynamics model through the following iterative equation:(12)$$\gamma_{i}\left(t\right)=K_{i}\int_{0}^{t}\left(-\hat{F}\left(\sigma \right)+\gamma_{i-1}\left(\sigma \right)\right)d\sigma ,\;i=2,\ldots ,r$$

Here, $\hat{F}=\gamma_{r}$, where r>0 is the order of the estimator. $\gamma_{1}\left(t\right)$ is obtained from Equation (Equation 10). When $F_{\mathrm{ext}}$, $\hat{F}$, and $\gamma_{i}\left(t\right)$ satisfy the Laplace transform conditions, the Laplace-transformed forms of Equations (Equation 11) and (Equation 12) are expressed as follows:(13)$$\begin{array}{cc} s\gamma_{1}\left(s\right) & =K_{1}\left(F_{\mathrm{ext}}\left(s\right)-\hat{F}\left(s\right)\right)\\ s\gamma_{i}\left(s\right) & =K_{i}\left(-\hat{F}\left(s\right)+\gamma_{i-1}\left(s\right)\right),\;i=2,\ldots ,r \end{array}$$
where *s* is the complex variable in the Laplace domain. The yields are rearranged as follows:(14)$$\hat{F}\left(s\right)=\gamma_{r}\left(s\right)=\frac{\prod_{i=1}^{r}K_{i}}{\sum_{i=0}^{r}s^{i}\prod_{j=i+1}^{r}K_{j}}F_{\mathrm{ext}}\left(s\right)$$

Here, matrix division refers to the element-wise division of corresponding diagonal elements. Thus, the estimated generalized external force is linearly related to the actual generalized external force in the Laplace domain, with the following transfer function:(15)$$G\left(s\right)=\frac{\prod_{i=1}^{r}K_{i}}{\sum_{i=0}^{r}s^{i}\prod_{j=i+1}^{r}K_{j}}=\frac{K_{1}K_{2}\cdots K_{r}}{s^{r}+K_{r}s^{r-1}+\cdots +K_{r}\cdots K_{2}s+K_{r}\cdots K_{1}}$$
where $G\left(s\right)\in C^{2n\times 2n}$ is a diagonal matrix. The *i*-th diagonal element of G(s) is expressed as follows:(16)$$G_{i}\left(s\right)=\frac{k_{1i}k_{2i}\cdots k_{ri}}{s^{r}+k_{ri}s^{r-1}+\cdots +k_{ri}\cdots k_{2i}s+k_{ri}\cdots k_{1i}}$$
where $k_{ji},j=1\cdots r$, is the *i*-th diagonal element of matrix $K_{j}$. $k_{1i}k_{2i}\cdots k_{ri}>0$ represents the gain of the observer, and when $\prod_{j}^{r}k_{ji},j=1\cdots r$ are coefficients of a Hurwitz polynomial, the stability of the estimator is guaranteed. The selection of *r* involves a trade-off between ideal estimation error and computational time, as a higher value of *r* results in increased computation time. The selection of $K_{i}$ is determined by two factors: ensuring the stability of Equation (Equation 16) and optimizing performance metrics such as overshoot, rise time, observation error, and other effects.

In practical implementation, the integrals in Equations (Equation 10) and (Equation 12) are discretized to enable real-time updates of $\hat{F}$. Based on the estimated generalized external force, the estimated external force acting on the feet can be derived as follows:(17)$$\hat{f}=J_{\mathrm{ext},j}^{T^{†}}\hat{F}$$
where $J_{\mathrm{ext},j}^{T^{†}}$ is the pseudo-inverse of $J_{\mathrm{ext},j}^{T}$.

## 3. Control Framework

This section introduces the robust disturbance rejection control framework for bipedal robots. The control framework consists of an high-level planner and a lower-level controller. The high-level planner utilizes the planner in our previous work [33]. The lower-level controller employs a weighted WBC scheme, which derives ground reaction forces for the momentum-based observer and compensates for the estimated disturbance effects on each task. The overall control scheme proposed in this paper is illustrated in Figure 2.

In our previous work [33], we proposed a balance and walking control scheme for bipedal robots, which utilized the same high-level planner as in this study but a different low-level controller. The low-level controller in the previous work employed a method that did not require solving inverse dynamics, enabling high control frequencies. This approach eliminated the dependency on a precise dynamic model and demonstrated rapid push recovery capabilities. In contrast, the current study differs in two key aspects: First, the focus has shifted to disturbance rejection, whereas the previous work primarily addressed basic standing and walking balance without considering disturbance rejection. Second, the low-level controller in this study adopts a whole-body control approach, which relies on a precise dynamic model and incorporates a momentum-based observer to enhance disturbance rejection performance. In comparison, the previous method did not utilize the robot’s precise dynamic model.

### 3.1. Footstep Planner

The high-level planner utilizes the MPC approach, employing a DCM-based LIPM for dynamics. The DCM-based LIPM dynamics simplify the original linear inverted pendulum model by introducing the divergent component of motion variable. Specifically, while the original LIPM dynamics are described by a second-order differential equation, the DCM-based formulation reduces them to two first-order differential equations. This simplification reveals that the DCM variable is inherently divergent in this model, necessitating its control to ensure system stability. In our paper, the planner simultaneously determines both the footstep location and the step duration to enhance walking robustness.

The equations of motion for the LIPM are given by the following:(18)$$\ddot{x}=\omega^{2}\left(x-p\right)$$
where x and $\ddot{x}$ are 2D vectors representing the horizontal position and acceleration of the CoM, respectively. The parameter $\omega =\sqrt{\frac{g}{z_{\mathrm{com}}}}$ is the reciprocal of the time constant of the LIPM, where *g* denotes gravitational acceleration and $z_{\mathrm{com}}$ is the fixed vertical position of the CoM. The variable p represents the horizontal position of the center of pressure (CoP), which is assumed to remain on the ground (i.e., its height is zero).

The DCM is defined as a linear combination of the CoM position and velocity:(19)$$\xi :=x+\frac{\dot{x}}{\omega }$$

By introducing DCM, the LIPM dynamics can be rewritten as two first-order linear differential equations:(20)$$\begin{array}{ccc} \dot{x} & = & \omega (\xi -x)\\ \dot{\xi } & = & \omega (\xi -p) \end{array}$$

The first equation in Equation (Equation 20) represents the stable component of the LIPM, where x converges to ξ. The second equation describes the unstable component, where ξ diverges along the line between ξ and p. Therefore, stabilizing ξ is necessary for achieving stable walking. Since this equation is relatively simple, we can derive its analytical solution:(21)$$\xi \left(t\right)=p_{0}+\left(\xi_{0}-p_{0}\right)e^{\omega t}$$
where $\xi_{0}$ and ξ(t) represent the DCM position at initial and at time *t*, respectively. Here, $p_{0}$ is assumed to be fixed during this period. With this solution, we can predict the future evolution of the DCM over multiple steps, which is essential for the MPC method.

The DCM offset is then defined as b=ξ−p, which represents the direction of the DCM velocity. The periodicity of the gait requires this offset to remain constant at the start and end of each step:(22)$$b=\xi_{0}-p_{0}=\xi_{T}-p_{T}$$
where *T* denotes the step duration.

Following the method introduced in [14], we can specify nominal step values, including nominal step length ($L_{nom}$), width ($W_{nom}$), duration ($T_{nom}$), and DCM offset ($b_{nom}={\left[b_{x,nom},b_{y,nom}\right]}^{T}$), which are related to desired average walking velocity. To eliminate the nonlinearity introduced by the step duration, we introduce a new decision variable $\eta =e^{\omega T}$ instead of directly using *T*.

The MPC method is employed to consider the changes in system variables over a future horizon, enabling the optimization of control effects over multiple steps. In this paper, our primary goal is to stabilize the DCM for the next few steps. Therefore, the cost function includes the error between the DCM offset and the nominal DCM offset at each step, the error between the step length and the nominal step length at each step, and the error between the step duration and the nominal step duration. Additionally, the solution must satisfy the DCM dynamics, including both intra-step and inter-step DCM dynamics, the kinematic constraints of the step length, and the constraints on the step duration. Ultimately, this forms a QP problem to solve for the optimal footstep positions for the next few steps and the optimal footstep timing for the current step. The decision variables of the QP problem include the footstep location, DCM offset of the next few steps, and the current footstep duration. Here, we set the foothold location as the CoP position:(23)$$\begin{array}{cccc} \; & \min_{p_{k},b_{k},\eta }\; & \; & \sum_{k=1}^{N_{s}}∥b_{k}-Rb_{nom}{∥}_{W_{b}}^{2}+∥l_{k}-Rl_{nom}{∥}_{W_{l}}^{2}+w_{\eta }{\left(\eta -e^{\omega T_{nom}}\right)}^{2}\\ \; & s.t.\; & \; & \xi_{1}=p_{0}+b_{0}e^{-\omega t_{0}}\eta \\ \; & \; & & \xi_{k}=p_{k-1}+b_{k-1}e^{\omega T_{nom}},\;k=2,\ldots ,N_{s}\\ \; & \; & & l_{min}\leq R^{T}l_{k}\leq l_{max},\;k=1,\ldots ,N_{s}\\ \; & \; & & e^{\omega T_{min}}\leq \eta \leq e^{\omega T_{max}} \end{array}$$
where $l_{nom}={\left[L_{nom},W_{nom}\right]}^{T}$ is the nominal step length in both the sagittal and coronal planes. The matrix R∈SO(2) accounts for foot yaw rotation. The parameter $t_{0}$ represents the elapsed time in the current step. The horizon length is denoted as $N_{s}$. $l_{min},l_{max}$ define the kinematic limits based on the robot’s range of motion. The parameters $T_{min}$ and $T_{max}$ limit the step duration and need to be tuned empirically. The matrices $W_{b},W_{l}$, and scalar weight $w_{\eta }$ are positive definite and serve as weighting factors for the DCM offset, step length, and step duration deviations in the cost function, respectively.

The main parameters of the MPC problem include the prediction horizon, the weights of the different components of the cost function, and the minimum and maximum step duration. Through extensive testing, we found that as the prediction horizon increased from one step to three steps, the walking stability improved. However, increasing it further to five steps did not yield additional benefits. At five steps, the computation time approaches the set frequency of 500 Hz. For the cost function weights, the DCM offset is prioritized with the largest weight, as it is the most critical for stability. Next is the nominal gait, and the smallest weight is assigned to the step duration. The setting of the maximum and minimum step duration is crucial for the walking performance of the BRUCE robot. Since the BRUCE robot is small (0.3 m in CoM height), as shown in Equation (Equation 21), its DCM diverges relatively quickly, requiring a shorter step duration. Thus, the maximum step duration is set to be smaller. At the same time, an excessively small step duration would cause the swing leg acceleration to become too large, which the robot’s motors cannot handle. Therefore, the maximum and minimum step durations are carefully adjusted based on the robustness of the bipedal walking performance.

After computing the optimal footstep location $p^{*}$ and timing $T^{*}$ through the above QP problem, the swing foot trajectory must be carefully designed to accommodate changing footstep location and timing. To ensure continuity up to the acceleration level, we employ a fifth-order polynomial as the reference trajectory:(24)$$p^{ref}\left(t\right)=\sum_{i=0}^{5}c_{i}t^{i},\;t_{0}\leq t\leq T^{*}$$
where $t_{0}$ represents the current time, measured from the beginning of the step.

### 3.2. Disturbance Rejection Whole-Body Control

This section employs the weighted WBC method from the literature [3], combined with the momentum-based observer from the previous section, to enhance its disturbance rejection capabilities. The weighted WBC does not require strict prioritization but instead allocates the importance of different tasks through weights in the cost function. By constructing a QP problem, the desired ground reaction forces and joint accelerations are solved. Subsequently, the feedforward torque is obtained through the whole-body dynamics, combined with joint-level Proportional and Derivative (PD) control to yield the final joint torques. Unlike other WBC methods based on full-body dynamics that do not consider disturbances, our approach integrates the full-body dynamics with estimated disturbances to compute more accurate ground reaction forces required for the observer, eliminating the dependency on foot-mounted sensors for ground reaction force measurement, which is a limitation of many momentum-based observer methods.

Recall that the external force estimated by the observer in previous section is $\hat{f}$, which can be divided into ${\hat{f}}_{\mathrm{st}}$ (estimated external force on the stance leg) and ${\hat{f}}_{\mathrm{sw}}$ (estimated external force on the swing leg). The external force on the stance leg primarily affects the control performance of the CoM, while the external force on the swing leg mainly impacts the tracking performance of the swing leg trajectory. Therefore, this paper incorporates the estimated external forces into the affected tasks for compensation, thereby achieving better task tracking performance under external disturbances.

The decision variables selected for the QP optimization in the whole-body control are $\zeta ={\left[{\ddot{r}}_{c}^{T}\;{\ddot{q}}^{T}\;f_{\mathrm{gr}}^{T}\right]}^{T}$, where ${\ddot{r}}_{c}={\left[{\ddot{x}}_{c}^{T}\;{\dot{\omega }}_{c}^{T}\right]}^{T}$. The cost function involves a weighted combination of multiple tracking task objectives. The main tasks are as follows:

#### 3.2.1. CoM Trajectory Tracking Task

First, the WBC method needs to track the planned CoM trajectory, which is derived from user input. When the desired CoM state, read from the user input, is $r_{c,\mathrm{ref}}$, ${\dot{r}}_{c,\mathrm{ref}}$, and ${\ddot{r}}_{c,\mathrm{ref}}$, the desired wrench at the CoM, $w_{\mathrm{com},\mathrm{des}}\in R^{6}$, can be computed using feedback and feedforward terms:(25)$$w_{\mathrm{com},\mathrm{des}}=K_{p,\mathrm{com}}\left(r_{c,\mathrm{ref}}-r_{c}\right)+K_{d,\mathrm{com}}\left({\dot{r}}_{c,\mathrm{ref}}-{\dot{r}}_{c}\right)+mg+M_{\mathrm{com}}\left(q\right){\ddot{r}}_{c,\mathrm{ref}}$$
where $K_{p,\mathrm{com}},K_{d,\mathrm{com}}\in R^{6\times 6}$ are the diagonal matrices containing PD gains for the CoM tracking task.

To achieve the desired wrench on CoM, we consider the first six rows of the whole-body dynamics equation (Equation (Equation 5)), which represent the dynamics of the CoM:(26)$$w_{\mathrm{com}}=M_{\mathrm{com}}\left(q\right)\left(\begin{array}{c} {\ddot{x}}_{c}\\ {\dot{\omega }}_{c} \end{array}\right)+\left[\begin{array}{c} mg\\ 0 \end{array}\right]=J_{\mathrm{st},\mathrm{com}}^{T}\left(q\right)f_{\mathrm{gr}}+J_{\mathrm{ext},\mathrm{com}}^{T}\left(q\right)f_{\mathrm{ext}}$$
where $w_{\mathrm{com}}\in R^{6}$ is the wrench at the robot’s CoM, including inertial and gravitational terms. From this equation, it is evident that the external force on the stance leg affects the control performance of the CoM trajectory. Therefore, the estimated external force on the support leg is incorporated into Equation (Equation 26), and the ground reaction force need to satisfy this equation. Thus, the cost function for the CoM tracking task is defined as follows:(27)$$L_{\mathrm{com}}={∥J_{\mathrm{st},\mathrm{com}}f_{\mathrm{gr}}+J_{\mathrm{st},\mathrm{com}}{\hat{f}}_{\mathrm{st}}-w_{\mathrm{com},\mathrm{des}}∥}_{W_{\mathrm{com}}}^{2}$$
where $W_{\mathrm{com}}$ is the weight matrix for the CoM trajectory tracking task, which can adjust the weights of the six degrees of freedom of the CoM to prioritize tracking in specific directions.

#### 3.2.2. Swing Foot Trajectory Tracking Task

Second, the WBC method need to track the planner swing foot trajectory to achieve stable walking. Given the planned swing foot position, velocity, and acceleration from the footstep planner, $p^{ref}$, ${\dot{p}}^{ref}$, and ${\ddot{p}}^{ref}$, respectively, the desired swing foot acceleration ${\ddot{p}}^{cmd}$ without considering external disturbances can be calculated using feedforward and feedback terms:(28)$${\ddot{p}}^{\mathrm{cmd}}={\ddot{p}}^{\mathrm{ref}}+K_{p,\mathrm{sw}}\left(p^{\mathrm{ref}}-x_{\mathrm{sw}}\right)+K_{d,\mathrm{sw}}\left({\dot{p}}_{\mathrm{ref}}-{\dot{x}}_{\mathrm{sw}}\right)$$
where $K_{p,\mathrm{sw}},K_{d,\mathrm{sw}}$ are the PD parameter matrices for the swing foot. $x_{\mathrm{sw}},{\dot{x}}_{\mathrm{sw}},$ denote swing foot position and velocity, respectively.

However, external disturbances on the swing leg affect the swing foot trajectory tracking performance. Therefore, the estimated external force on the swing leg ${\hat{f}}_{\mathrm{sw}}$ is used to compensate for the disturbance. To achieve this, the relationship between the swing foot acceleration and the external force must be established. Considering the whole-body dynamics, the orthogonal null space projector $P\in R^{6+2n\times 6+2n}$, which satisfies $PJ_{\mathrm{st}}^{T}=0$ and $P^{2}=P^{T}=P$, is used. The projection matrix P eliminates the contact constraint-related term $J_{\mathrm{st}}^{T}f_{\mathrm{gr}}$ in the dynamics equation [34]. Following the method in [35,36], P is selected as $P=I_{6+2n}-J_{\mathrm{st}}^{+}J_{\mathrm{st}}$. Multiplying both sides of Equation (Equation 5) by P yields the following:(29)$$P\left(M\dot{v}+h\right)=PS^{T}\tau +P{\left[\begin{array}{c} J_{\mathrm{sw}}\\ J_{\mathrm{st}} \end{array}\right]}^{T}\left[\begin{array}{c} f_{\mathrm{sw},\mathrm{ext}}\\ f_{\mathrm{st},\mathrm{ext}} \end{array}\right]=PS^{T}\tau +PJ_{\mathrm{sw}}^{T}f_{\mathrm{sw},\mathrm{ext}}$$
where h is the sum of Coriolis and gravity force. Since $PJ_{\mathrm{st}}^{T}=0$, the only term related to external forces is $PJ_{\mathrm{sw}}^{T}f_{\mathrm{sw},\mathrm{ext}}$, which is associated with the external force on the swing leg. Following [36], the equation can be transformed into the following:(30)$$M_{c}\dot{v}+Ph-Cv=PS^{T}\tau +PJ_{\mathrm{sw}}^{T}f_{\mathrm{sw},\mathrm{ext}}$$
where $M_{c}=PM+I_{6+2n}-P$ and $C=-J_{\mathrm{st}}^{+}J_{\mathrm{st}}$. As long as M is invertible, $M_{c}$ is also invertible.

Based on the floating base kinematics, the relationship between the generalized coordinates and the swing foot’s velocity and acceleration is as follows:(31)$$\begin{array}{cc} {\dot{x}}_{\mathrm{sw}} & =J_{\mathrm{sw}}v\\ {\ddot{x}}_{\mathrm{sw}} & =J_{\mathrm{sw}}\dot{v}+{\dot{J}}_{\mathrm{sw}}v \end{array}$$

Multiplying both sides of Equation (Equation 30) by $M_{c}^{-1}$ yields the relationship for $\dot{v}$, which is substituted into Equation (Equation 31) to obtain the relationship between the swing foot acceleration and the external force:(32)$${\ddot{x}}_{\mathrm{sw}}=J_{\mathrm{sw}}M_{c}^{-1}PS^{T}\tau +J_{\mathrm{sw}}M_{c}^{-1}PJ_{\mathrm{sw}}^{T}f_{\mathrm{sw},\mathrm{ext}}+{\dot{J}}_{\mathrm{sw}}v-J_{\mathrm{sw}}M_{c}^{-1}\left(Ph-Cv\right)$$

To mitigate the disturbance on swing leg, we can compensate for the effect of the external force on the swing foot acceleration, i.e., the second term in the equation above. Therefore, the desired swing foot acceleration after compensation becomes the following:(33)$${\ddot{p}}^{\mathrm{des}}={\ddot{p}}^{\mathrm{cmd}}-J_{\mathrm{sw}}M_{c}^{-1}PJ_{\mathrm{sw}}^{T}{\hat{f}}_{\mathrm{sw}}$$

The desired swing foot acceleration is realized through the kinematic relationship in Equation (Equation 31), resulting in the cost function for the swing foot trajectory tracking task with disturbance compensation:(34)$$L_{\mathrm{sw}}={∥J_{\mathrm{sw}}\dot{v}+{\dot{J}}_{\mathrm{sw}}v-{\ddot{p}}^{\mathrm{des}}∥}_{W_{sw}}^{2}$$
where $W_{sw}$ is the weight matrix for the swing foot trajectory tracking task.

#### 3.2.3. Stance Foot Contact Unmoving Task

This task ensures that the stance foot remains fixed at its contact point with the ground, preventing any relative motion such as slipping or lifting. To achieve this, we enforce a zero relative velocity constraint at the contact point, ensuring that it remains stationary with respect to the ground. This condition is formulated similarly to the kinematic equation for the swing foot position (Equation 31), leading to the following:(35)$$\begin{array}{cc} 0_{3n_{c}} & =J_{\mathrm{st},\mathrm{com}}\left(q\right){\dot{r}}_{c}+J_{\mathrm{st},j}\left(q\right)\dot{q}\\ 0_{3n_{c}} & =J_{\mathrm{st},\mathrm{com}}\left(q\right){\ddot{r}}_{c}+{\dot{J}}_{\mathrm{st},\mathrm{com}}\left(q,\dot{q}\right){\dot{r}}_{c}+J_{\mathrm{st},j}\left(q\right)\ddot{q}+{\dot{J}}_{\mathrm{st},j}\left(q,\dot{q}\right)\dot{q} \end{array}$$

Based on the above equation, the cost function for the stance foot contact unmoving task is as follows:(36)$$L_{\mathrm{st}}={∥J_{\mathrm{st},\mathrm{com}}\left(q\right){\ddot{r}}_{c}+{\dot{J}}_{\mathrm{st},\mathrm{com}}\left(q,\dot{q}\right){\dot{r}}_{c}+J_{\mathrm{st},j}\left(q\right)\ddot{q}+{\dot{J}}_{\mathrm{st},j}\left(q,\dot{q}\right)\dot{q}-0∥}_{W_{\mathrm{st}}}^{2}$$
where $W_{st}$ is the weight matrix for the stance foot contact unmoving task. Maintaining contact point unmoving is typically a constraint, but in this paper, it is treated as a task-space objective, i.e., a soft constraint. This approach often speeds up the QP and provides better numerical stability [37]. With sufficient task weights, the effect of null space projection can be achieved, ensuring that other tasks comply with the non-slip contact condition.

In addition to the task-related cost functions, the optimization also minimizes the generalized coordinate acceleration and ground contact force, both of which are included in the decision variables. The corresponding cost function is ${∥\zeta ∥}_{R}^{2}$.

#### 3.2.4. Constraints

For the constraint equations, the first is the motion equation, which restricts the motion to satisfy the first six rows of the floating base dynamics equation (Equation (Equation 5)):(37)$$\left[M_{\mathrm{com}}\left(q\right)\;0_{6\times 2n}\;-J_{\mathrm{st},\mathrm{com}}^{T}\left(q\right)\right]\zeta =\left[\begin{array}{c} -mg\\ 0 \end{array}\right]+J_{\mathrm{ext},\mathrm{com}}^{T}\left(q\right)f_{\mathrm{ext}}$$

The second constraint ensures that the ground reaction forces satisfy the friction cone constraints to avoid slipping. To avoid the nonlinearity of this constraint, the friction cone is approximated as a square pyramid, resulting in linear constraints in the optimization problem. The ground reaction force $f_{\mathrm{gr},i}\in R^{3}$, $i=1,\ldots ,n_{c}$, satisfies the contact constraint with friction coefficient μ as follows:(38)$$\underset{U}{\underset{︸}{\left[\begin{array}{ccc} \pm 1 & 0 & -\mu \\ 0 & \pm 1 & -\mu \\ 0 & 0 & -1\\ 0 & 0 & 1 \end{array}\right]}}f_{\mathrm{gr},i}\leq \underset{b}{\underset{︸}{\left[\begin{array}{c} 0\\ 0\\ -f_{z,min}\\ f_{z,max} \end{array}\right]}}$$
where $f_{z,min},f_{z,max}$ are the minimum and maximum normal forces to constrain the magnitude of the contact force. Note that a positive minimum value prevents contact detachment.

The final QP formulation of our disturbance rejection WBC method is as follows:(39)$$\begin{array}{cc} \underset{\zeta =[{\ddot{r}}_{c}\;\ddot{q}\;f_{\mathrm{gr}}]}{\mathrm{minimize}}\; & L_{\mathrm{com}}+L_{\mathrm{sw}}+L_{\mathrm{st}}+{∥\zeta ∥}_{R}^{2}\\ \mathrm{subject}\;\mathrm{to}\; & \left[M_{\mathrm{com}}\left(q\right)\;0_{6\times 2n}\;-J_{\mathrm{st},\mathrm{com}}^{T}\left(q\right)\right]\zeta =\left[\begin{array}{c} -mg\\ 0 \end{array}\right]+J_{\mathrm{ext},\mathrm{com}}^{T}\left(q\right)f_{\mathrm{ext}}\\ \; & Uf_{\mathrm{gr},i}\leq b\;i=1,\ldots ,n_{c} \end{array}$$

The main parameters of our WBC method include the PD parameters for the CoM trajectory and swing leg trajectory tracking tasks, the weight matrix for each task and the minimum and maximum normal forces. For the PD gains, the tuning process relies on trial and error, guided by tracking performance metrics such as the CoM velocity tracking error and swing leg tracking error. For the weight matrix, the highest weight is assigned to the stance foot contact unmoving task, as it serves as a hard constraint. The CoM task is assigned the next highest weight, with maintaining the CoM height being the most critical among its directional components. The lowest weight is assigned to the swing foot trajectory tracking task. For the minimum and maximum normal forces, a small minimum normal force is set to prevent contact detachment. The maximum normal force is chosen to avoid generating ground reaction forces that exceed the motor’s capabilities.

After obtaining the optimal $\zeta^{*}=\left[{\ddot{r}}_{c}^{*}\;{\ddot{q}}^{*}\;f_{\mathrm{gr}}^{*}\right]$, the feedforward torque $\tau_{\mathrm{ff}}$ is obtained by substituting into the second row of the Equation (Equation 5):(40)$$\tau_{\mathrm{ff}}=M_{q}{\ddot{q}}^{*}+C\left(q,v\right)\dot{q}-J_{st,j}^{T}\left(q\right)f_{\mathrm{gr}}^{*}-J_{\mathrm{ext},j}^{T}f_{\mathrm{ext}}$$

The joint torque command is obtained by adding feedforward and joint position feedback $\tau_{\mathrm{fd}}$ together as follows:(41)$$\tau_{cmd}=\tau_{\mathrm{ff}}+k_{p}\left(q^{\mathrm{des}}-q\right)+k_{d}\left({\dot{q}}^{\mathrm{des}}-\dot{q}\right),\;\tau_{\min}\leq \tau_{\mathrm{cmd}}\leq \tau_{\max}$$
where $k_{p},k_{d}$ are positive-definite diagonal matrices containing PD feedback gains. $q^{\mathrm{des}},{\dot{q}}^{\mathrm{des}}$ are the desired joint position and velocity obtained from inverse kinematics. $\tau_{\min},\tau_{\max}$ are the minimum and maximum joint torques, respectively.

## 4. Simulations and Experiments

In this section, a series of simulations and real-world experiments were conducted to validate the effectiveness of our controller. Both the simulations using MuJoCo (Version 3.1.4) [38] and experiments were tested with a miniature bipedal robot, BRUCE. Each leg of BRUCE has five degrees of freedom, including a spherical hip joint, a knee joint, and an ankle joint. The robot’s state estimation is supported by an onboard IMU, joint encoders, and contact-sensing feet. A mini PC with an Intel Core i5-7260U Dual-Core CPU at 2.2 GHz was utilized as the onboard computing resource.

The 3D model of the BRUCE robot employed in our simulations is based on the model made available by the official developers [39], with the necessary sensors added to the original model files. The QP problems in both the footstep planner and the controller were solved using the OSQP solver (Version 0.6.5) [40]. With the onboard PC and OSQP, both our control scheme and momentum-based observer operate at an update frequency of 500 Hz.

The software architecture is designed to operate in a multithreaded environment, enabling the simultaneous execution of all modules in the control block diagram depicted in Figure 2. The architecture comprises six primary threads: a motor command thread operating at 1000 Hz, a state estimation thread at 500 Hz, a user input thread at 100 Hz, a momentum-based observer thread at 500 Hz, a high-level planner thread at 500 Hz, and a low-level whole-body control thread at 500 Hz. Data communication utilizes a shared memory based on Python module posix_ipc [41]. The program is developed in Python 3.10, with Numba [42] employed for pre-compilation to optimize the computational speed of state estimation, kinematics, and dynamics calculations. The data curves are generated by recording the necessary data using variables during the software execution, saving it into a CSV file, and then processing and plotting the data using MATLAB R2024a.

### 4.1. Simulations

To achieve better walking disturbance rejection, the control parameters were tuned through many attempts. The final parameters used in the simulation are as follows: In the high-level planner, the weight matrices are set to $W_{b}=\mathrm{diag}\left(100,100\right)$, $W_{l}=\mathrm{diag}\left(1,1\right)$, $w_{\eta }=0.1$, and the prediction horizon $N_{s}$ is 3. The maximum and minimum step durations are set to $T_{max}=0.3$ and $T_{min}=0.18$, respectively. For the CoM trajectory tracking task in the WBC, the gain parameters for the desired wrench are $K_{p,com}=\mathrm{diag}\left(100,100,900,250,250,100\right)$ and $K_{d,com}=\mathrm{diag}\left(50,50,120,100,100,50\right)$. For the swing foot trajectory tracking task, the desired acceleration gains are $K_{p,\mathrm{sw}}=100I_{5}$ and $K_{d,\mathrm{sw}}=25I_{5}$. Since the swing leg has only five degrees of freedom, it is not possible to control the 3D positions of both the toe and heel of the swing foot simultaneously. Therefore, the control of the x-direction of the heel is omitted. The weights in the QP problem are set to $W_{\mathrm{com}}=\mathrm{diag}\left(5,5,100,5,5,5\right)$, $W_{\mathrm{sw}}=10I_{5}$, $W_{\mathrm{st}}=10^{3}I_{6}$, and R=I.

An analysis of the observer order was conducted to compare the estimation errors of observers with different orders. Figure 3 illustrates the estimation errors over time when a constant external force is applied to the robot. The first-order observer exhibits a higher overshoot in error compared to the second-order observer. As the order increases, the overshoot decreases. However, the overshoot increases again at the fourth order. If the order is further increased, the computation time becomes excessively long and there would be too many parameters, so no further comparisons were made. Among these four orders, the third-order observer performs the best. Therefore, in subsequent experiments, the order *r* was set to 3, and the coefficients $K_{1}$, $K_{2}$, and $K_{3}$ were chosen as $0.02I_{10}$, $0.42I_{10}$, and $19I_{10}$, respectively.

#### 4.1.1. Stable Walking Under External Disturbance on the Swing Leg

To verify the improvement in disturbance rejection performance of our WBC method, we first tested the standard WBC approach without the momentum-based observer [3], which serves as the baseline controller provided by the developers of the BRUCE robot. During the test, the robot was subjected to external disturbances applied to the swing leg. As shown in Figure 4, when a forward external force of 5 N is applied to the swing leg during walking, the swing leg deviates from the planned foothold, causing the robot to lose stability after two or three steps.

When the proposed disturbance rejection WBC method is employed, the observer provides an estimate of the external force under the same conditions, as shown in Figure 5. It can be seen from the figure that the observer can quickly estimate the magnitude of the external force, and the estimation speed meets the requirements of the control method. With the momentum-based observer, the deviation of the swing leg caused by the external force is quickly compensated, allowing it to return to the planned foothold. The 3D spatial position error curves between the desired and actual positions of the right foot are shown in Figure 6. It can be observed that the swing foot is subjected to an external force at around 3 s, as shown in Figure 5. At this point, the swing foot error begins to increase, reaching a maximum deviation of 2 cm. As the estimation of the external force becomes more accurate, the disturbance is compensated, causing the error to decrease rapidly and allowing the swing foot to realign with the planned trajectory. This simulation verifies that under the same external disturbance, the standard WBC approach loses stability, while the proposed disturbance rejection WBC method maintains stable walking.

#### 4.1.2. Stable Walking on Uneven Terrain

To further validate the performance of the proposed control scheme, we tested its effectiveness in controlling the BRUCE robot walking on uneven terrain. Comparisons were made with the standard WBC method and the CoM-based disturbance rejection WBC method from the literature [25], which only compensates for disturbances acting on the CoM. The uneven terrain consists of multiple 1 cm high wooden boards irregularly distributed across the area. Screenshots of the simulations for the three control methods are shown in Figure 7. Both the proposed disturbance rejection WBC method, which considers disturbances on both legs, and the CoM-based disturbance rejection WBC method enabled the robot to traverse the entire uneven terrain, whereas the standard WBC method failed.

The CoM trajectories of the robot walking on uneven terrain using CoM-based disturbance rejection WBC method and the proposed method are shown in Figure 8. From the CoM curve in the *x*-direction, it can be seen that the proposed method maintains a consistent slope during walking, with smaller velocity fluctuations. In the *y*-direction, although there is no walking velocity command in the *y*-direction, both methods inevitably experience deviations in the *y*-direction because of uneven terrain. However, compared to the CoM-based method, the proposed method exhibits smaller deviations. The simulation results demonstrate that the proposed disturbance rejection WBC method, compared to the CoM-based method, results in smaller CoM deviation errors and better trajectory tracking performance under such disturbances. Moreover, the proposed method enhances the robot’s robustness, enabling it to traverse such uneven terrain, unlike the standard WBC method.

### 4.2. Experiment

To evaluate the practicality of the proposed control scheme in real world environment, we conducted experimental validation on the biped robot BRUCE. The parameters used in the planning method for the experiments are same as those used in the simulations. For the CoM trajectory tracking task in the WBC, the gain parameters for the desired wrench are $K_{p,com}=\mathrm{diag}\left(100,100,800,200,200,100\right)$ and $K_{d,com}=\mathrm{diag}\left(50,50,150,100,100,50\right)$. For the swing foot trajectory tracking task, the desired acceleration gains are $K_{p,\mathrm{sw}}=100I_{5}$ and $K_{d,\mathrm{sw}}=10I_{5}$.

#### 4.2.1. Walking Experiment Under External Swing Leg Disturbance

In this experiment, an external disturbance was applied to the robot’s swing leg using a tether attached to the right leg. The other end of the tether was connected to a force sensor to measure the magnitude and duration of the applied force. Figure 9 shows snapshots of the experiment. Initially, the tether was slack, and no external force was applied to the swing leg. When the tether was tightened, the swing leg began to experience an external force. Once the swing leg touched the ground, the external force was removed, and the tether returned to a slack state. From the figure, it can be seen that the robot maintains stable walking even when the swing leg is subjected to an external disturbance. The experiment demonstrated that the robot could withstand a maximum external force of 5 N while maintaining stability, with the disturbance lasting approximately 0.2 s.

Figure 10 depicts the deviation curve between the actual and desired positions of the swing foot during the experiment. Here, the deviation is the 3D spatial distance deviation *e*, calculated as the square root of the sum of the squared deviations in the three spatial directions $e_{x}$, $e_{y}$, and $e_{z}$, i.e., $e=\sqrt{e_{x}^{2}+e_{y}^{2}+e_{z}^{2}}$. As shown in the figure, when no external force was applied, the deviation between the actual and target positions remained within 2 cm. At approximately 1.1 s, the swing foot began to experience an external force, causing the position deviation to increase to 3.5 cm within 0.1 s. Due to the observer’s estimation and compensation of the external force in the proposed control method, the deviation decreased to 1.6 cm within 0.2 s, returning to the range observed in the absence of disturbance. This reduction minimized the impact of the external force on the swing foot’s landing position. Furthermore, the subsequent deviation remained within the 2 cm range, indicating that the effect of the external disturbance has been compensated and did not produce any further impact. The experimental results demonstrate that the proposed disturbance rejection control method effectively reduces the position deviation of the swing foot, ensuring stable walking of the robot when swing leg is under disturbance.

#### 4.2.2. Walking Experiment Under Complex Disturbance

To further evaluate the disturbance rejection performance of the proposed method, an experiment was conducted to simultaneously resist multiple types of disturbances, including those from uneven terrain and external forces. Uneven terrain was created by placing wooden boards of varying heights (5 mm and 10 mm) on the ground. External forces were applied to the robot’s upper body using a stick with a force sensor at its tip. Figure 11 shows snapshots of this experiment. From the third subfigure, it can be seen that the BRUCE robot’s left foot is stepping on a 10 mm wooden board, while the right foot is on flat ground, and the upper body is subjected to an external force. The results demonstrate that the robot maintained stable walking even under these complex disturbances. During the experiment, the external force applied to the upper body was 10 N, lasting approximately 0.7 s.

Figure 12 illustrates the trajectories of the CoM, left foot, right foot, and divergent component of motion in the *x*-direction during the experiment. It can be observed that at the beginning of the experiment, the robot walked on uneven terrain at a speed of approximately 0.2 m/s. At around 1 m (approximately 3.2 s), the upper body began to experience an external force. Under the influence of this force, the CoM position stopped moving forward. To maintain stability, the right foot moved backward by about 10 cm. After the external force was removed, the robot quickly recovered to a walking speed of 0.2 m/s. The experimental results demonstrate that the proposed disturbance rejection control method enables the robot to maintain stable walking while simultaneously resisting disturbances from uneven terrain and external forces.

## 5. Discussion and Conclusions

In this paper, we presented a complete disturbance rejection control scheme for bipedal robots. The high-level DCM-based planner selects the desired foothold position and step duration using the MPC method, enhancing gait robustness through adaptive step duration adjustments. The low-level control method integrates a momentum-based observer, capable of estimating disturbances on both stance and swing legs, and a weighted WBC method based on full-body dynamics with estimated disturbances. Our method compensates for disturbance on both stance leg and swing leg, enabling the robot to adapt to complex situations. Notably, our method does not require foot-mounted sensors to measure ground reaction forces, making it applicable to a wider range of bipedal robots. We conducted a series of simulations and experiments to demonstrate the effectiveness of our control scheme. Compared to the standard WBC method, our disturbance rejection WBC method effectively handles disturbances on the swing leg. In comparison to the CoM-based disturbance rejection WBC method, our method results in smaller CoM deviation errors and better trajectory tracking performance when controlling the robot’s walking on uneven terrain.

## Figures and Tables

**Figure 1 biomimetics-10-00189-f001:**
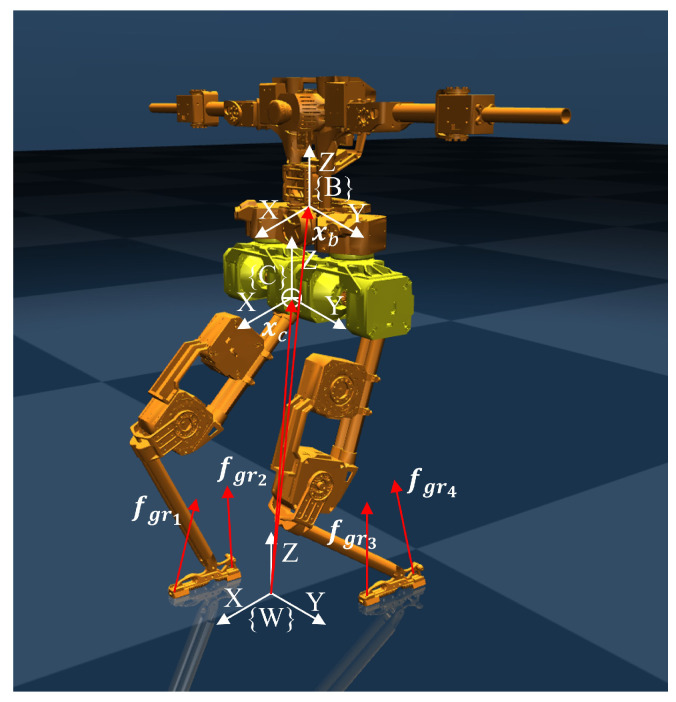
Overview of the coordinate systems employed in the formulation of robot’s dynamics.

**Figure 2 biomimetics-10-00189-f002:**
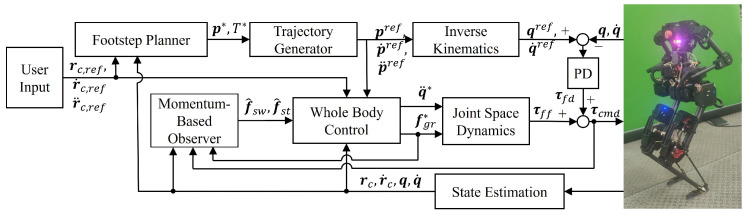
Block diagram of the control framework.

**Figure 3 biomimetics-10-00189-f003:**
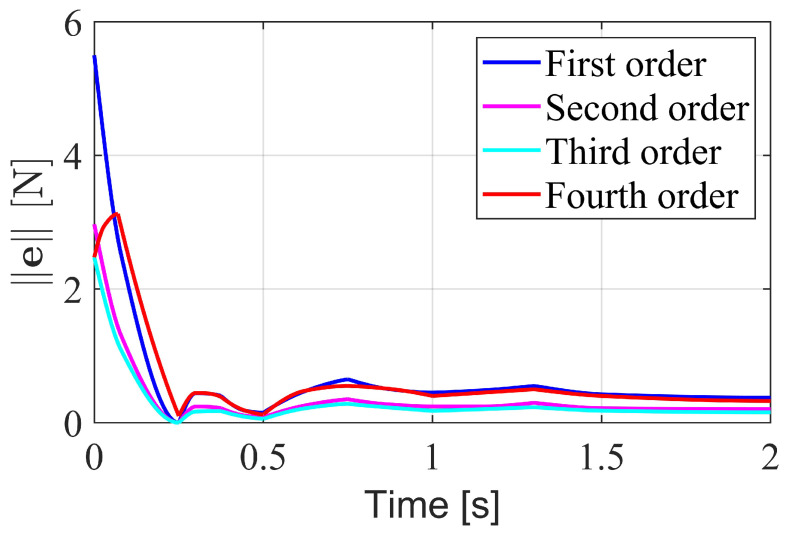
Estimation errors of observers with different orders.

**Figure 4 biomimetics-10-00189-f004:**
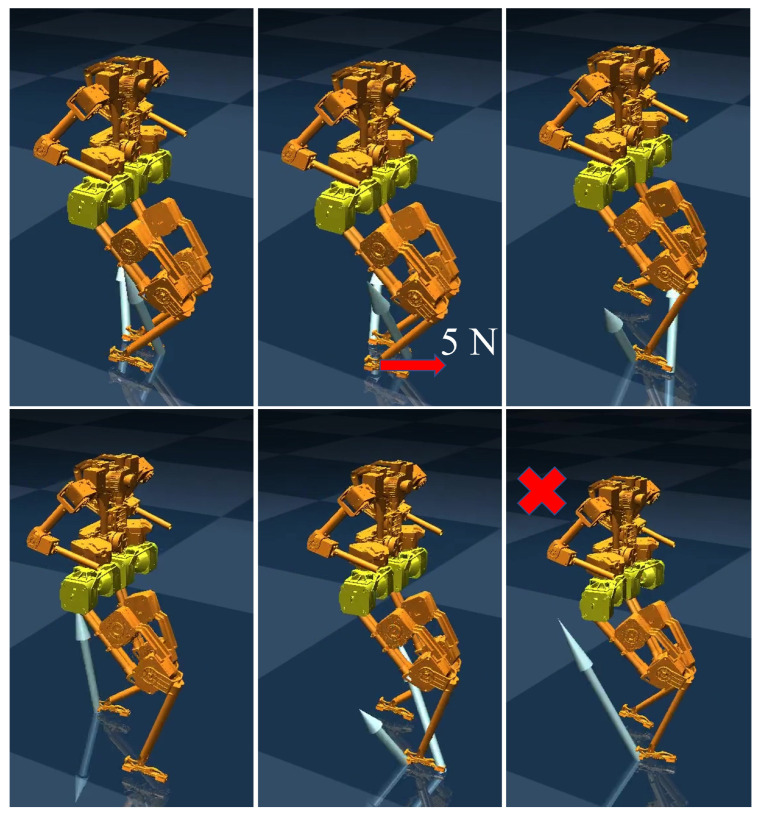
Screenshot of the standard WBC’s swing leg disturbance rejection simulation. The ‘x’ marker denotes the robot’s falling state.

**Figure 5 biomimetics-10-00189-f005:**
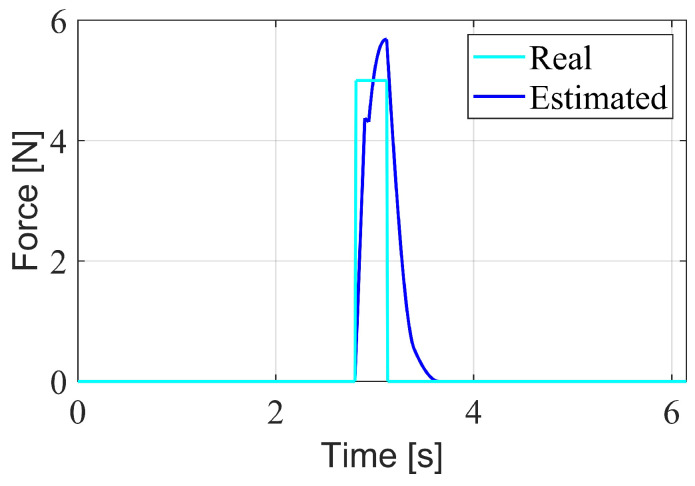
External force estimated by the momentum-based observer.

**Figure 6 biomimetics-10-00189-f006:**
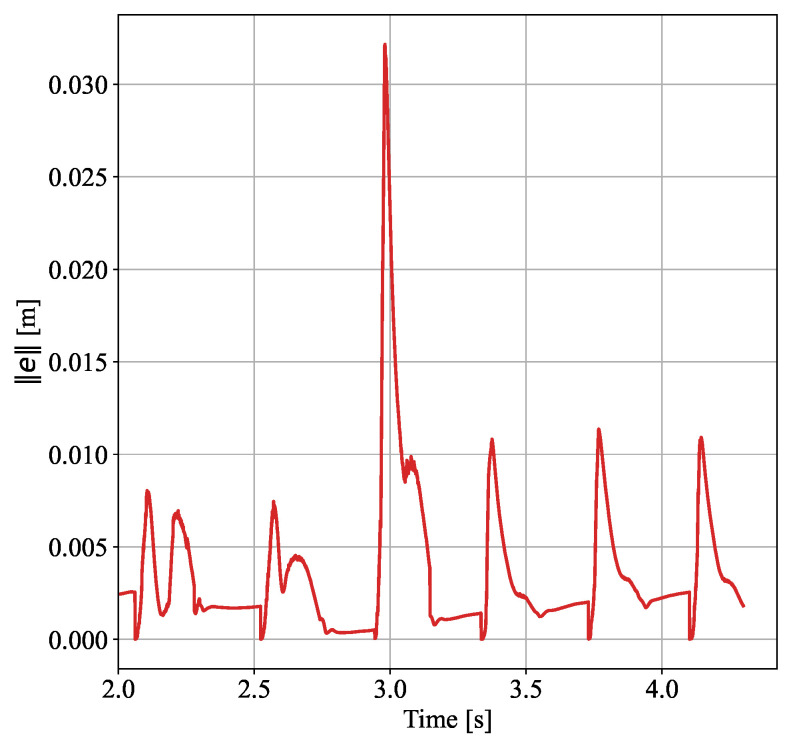
Error between actual and desired right foot position.

**Figure 7 biomimetics-10-00189-f007:**
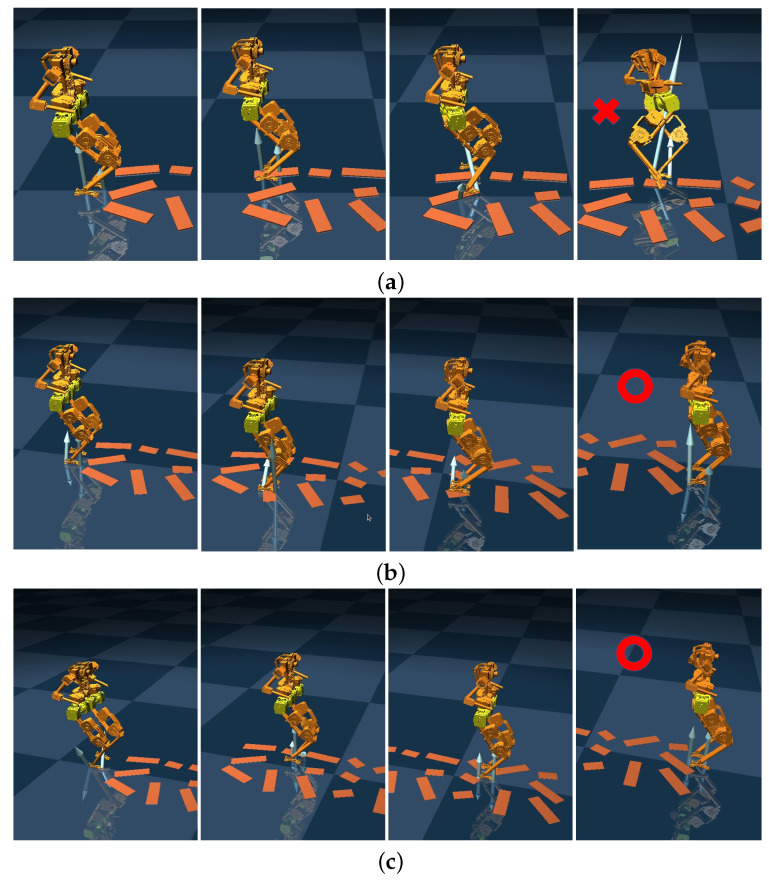
Simulation results of walking on uneven ground under three different control methods. The ‘x’ denotes falling incidents during locomotion, whereas ‘o’ represents successful navigation across the uneven terrain. (**a**) Standard WBC method; (**b**) CoM-based disturbance rejection WBC method in paper [25]; (**c**) proposed disturbance rejection WBC method considering disturbances on both legs.

**Figure 8 biomimetics-10-00189-f008:**
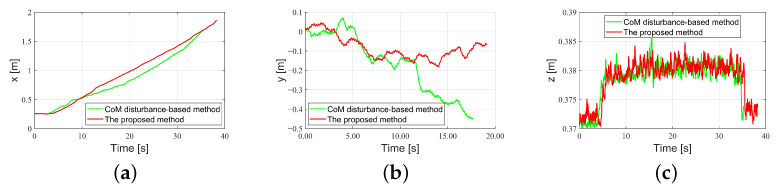
Comparison of CoM trajectories between the CoM-based disturbance rejection WBC method and the proposed method. (**a**) *x* direction; (**b**) *y* direction; (**c**) *z* direction.

**Figure 9 biomimetics-10-00189-f009:**
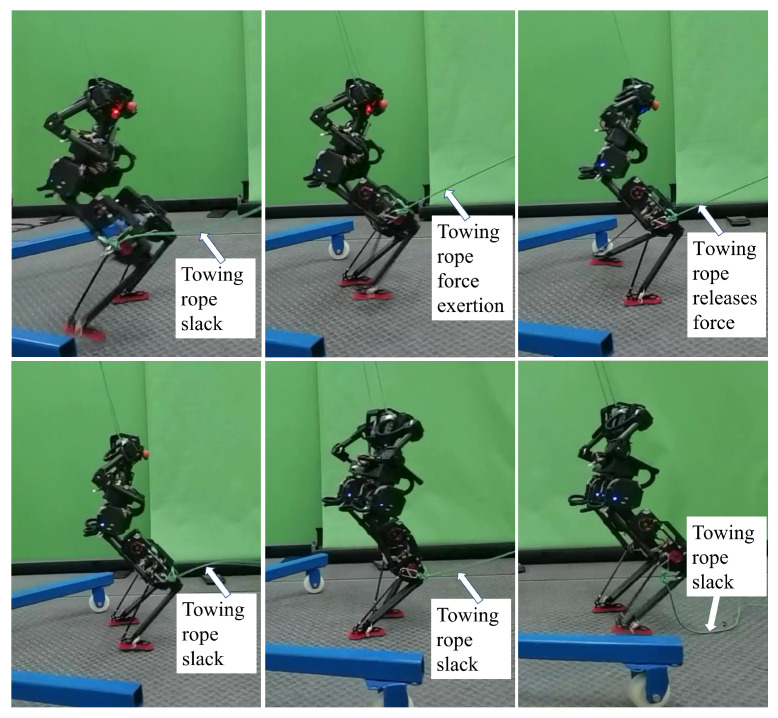
Snapshot of the swing leg disturbance experiment.

**Figure 10 biomimetics-10-00189-f010:**
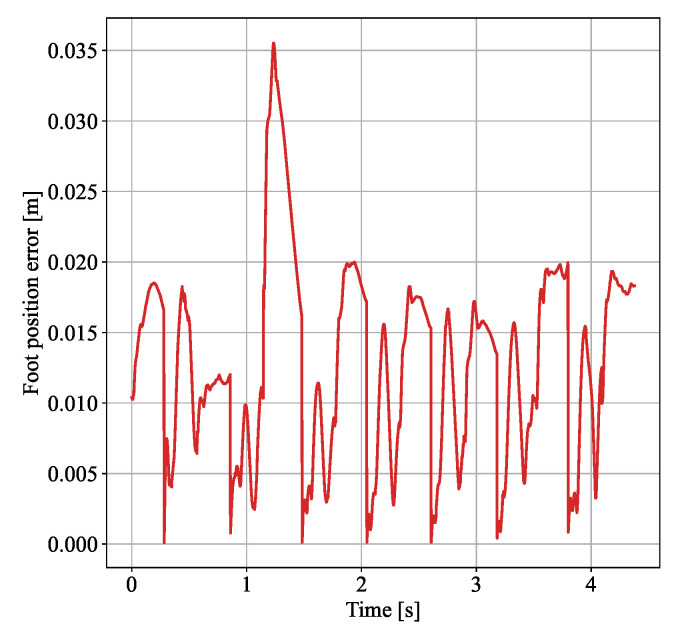
Swing foot position deviation during swing leg disturbance experiment.

**Figure 11 biomimetics-10-00189-f011:**
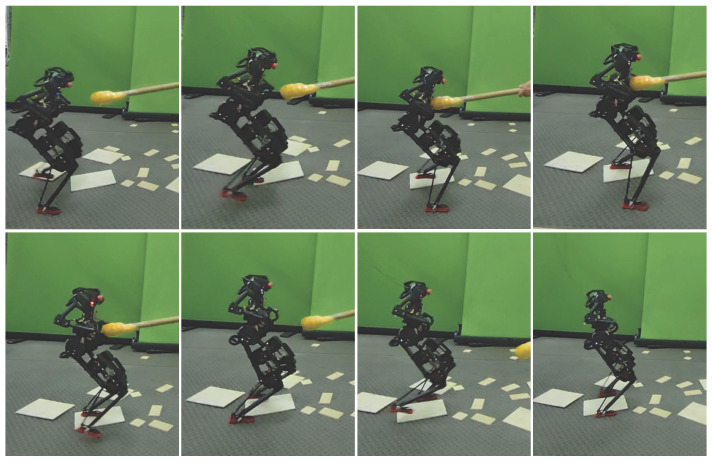
Snapshots of the experiment under complex disturbance conditions.

**Figure 12 biomimetics-10-00189-f012:**
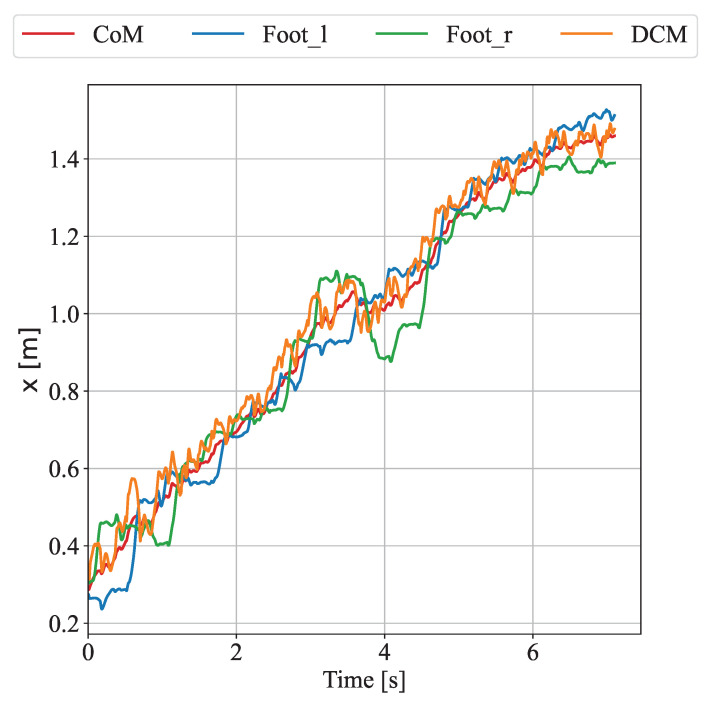
Trajectories of the CoM, left foot, right foot, and DCM in the *x*-direction during the complex disturbance experiment.

## Data Availability

The data and simulation code used to support the findings of this study are available from Shuai Heng upon request.

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
