# Peer review of "A Robust Disturbance Rejection Whole-Body Control Framework for Bipedal Robots Using a Momentum-Based Observer"

_biomimetics, 2025, doi:10.3390/biomimetics10030189_

Round 1
Reviewer 1 Report
Comments and Suggestions for Authors
I appreciate the author for the work and the paper is well-written.
Author Response
C1-1:”I appreciate the author for the work and the paper is well-written.”
Response: We appreciate the reviewer’s kind words regarding our
work and the clarity of the manuscript. Your positive feedback is
truly encouraging, and we are grateful for the time and effort you
have dedicated to reviewing our paper.
Reviewer 2 Report
Comments and Suggestions for Authors
The authors have prepared a paper that may be of interest to researchers developing control systems for walking robots. There are some comments that the authors should consider:
1. The authors should describe in detail the comparison of this paper with their previous paper [32]: what methods and results were obtained in [32] and how the new methods differ from the previous ones.
2. The authors should explain how exactly the software implementation of the developed controller is organized. The graph in Figure 3 looks like it was obtained in MATLAB. If the authors obtained the graph there, then how did they organize the connection between MuJoCo and MATLAB? Was the 3D model of the BRUCE robot developed by the authors? Or where did they get it from? What does the control scheme look like in the software environment? The description given in the article is not enough for reproducibility of the research.
3. Does the real BRUCE robot have the original basic controller developed by the manufacturer? If so, how quantitatively is the control improved in comparison with the basic controller?
Minor comments:
-why don't the authors introduce notations for some variables? for example, calculated in (27), (34), (36)
-in some places "where" after formulas is written with a capital letter (for example, line 312) or with an indent (for example, line 248, 259, 302)
Reviewer 3 Report
Comments and Suggestions for Authors
This is an interesting work with interesting results.
However, there are some points to be clarified. They are indicated below:
- Line 112: What is a virtual joint?
- Lines 110-112: What is the exact difference between the frames {C} and {B}?
- Lines 122-124: What kind of an Euler angle sequence is used in defining the orientation matrices, angular velocities, and angular accelerations?
- Lines 150-153: The motion definitions are not very clear. In particular, what is the internal motion of a multi-body system?
- Equation (6): Here, the momentum is denoted by the symbol a. However, this is an improper and unfortunate notation, because a is the almost universal symbol reserved for acceleration.
- Equation (12): This equation is introduced as an extension of Equation (11). What is the motivation and justification of this extension? What are the criteria in choosing the number of terms (r) and the gain Ki for the i'th term?
- Section 3. Control Framework: The methods and formulations relevant to all of the Subsections (3.1, 3.2.1, 3.2.2, 3.2.3, and 3.2.4) of this section are not explained very clearly. Especially concerning Subsection 3.1, the description of the DCM-based LIPM dynamics and the application of the MPC method with its selected parameters must be explained clearly with sufficient detail.
Except a very few grammatical mistakes, the quality of English language looks fine.
However, the relevant concepts and formulations could be explained more clearly.
Moreover, unless readily understandable, the usage of acronyms had better be minimized.
Round 2
Reviewer 3 Report
Comments and Suggestions for Authors
The authors made the suggested amendments.
So, the paper can be published according to my opinion.